# Limonene Exerts Anti-Inflammatory Effect on LPS-Induced Jejunal Injury in Mice by Inhibiting NF-κB/AP-1 Pathway

**DOI:** 10.3390/biom14030334

**Published:** 2024-03-12

**Authors:** Sarmed H. Kathem, Yasameen Sh. Nasrawi, Shihab H. Mutlag, Surya M. Nauli

**Affiliations:** 1Pharmacology and Toxicology Department, College of Pharmacy, University of Baghdad, Baghdad 92618, Iraq; shihabhattab@copharm.uobaghdad.edu.iq; 2Department of Pharmacy, Al-Zahrawi University College, Karbala 56001, Iraq; yasmin.jawad2100m@copharm.uobaghdad.edu.iq; 3Biomedical and Pharmaceutical Sciences Department, School of Pharmacy, Chapman University, Irvine, CA 92866, USA; nauli@chapman.edu

**Keywords:** TNF-α, IL-1β, iNOS, IRF3, TLR4, intestinal injury, monoterpenes

## Abstract

The human gastrointestinal system is a complex ecosystem crucial for well-being. During sepsis-induced gut injury, the integrity of the intestinal barrier can be compromised. Lipopolysaccharide (LPS), an endotoxin from Gram-negative bacteria, disrupts the intestinal barrier, contributing to inflammation and various dysfunctions. The current study explores the protective effects of limonene, a natural compound with diverse biological properties, against LPS-induced jejunal injury in mice. Oral administration of limonene at dosages of 100 and 200 mg/kg was used in the LPS mouse model. The Murine Sepsis Score (MSS) was utilized to evaluate the severity of sepsis, while serum levels of urea and creatinine served as indicators of renal function. Our results indicated that LPS injection induced renal function deterioration, evidenced by elevated serum urea and creatinine levels compared to control mice. However, pretreatment with limonene at doses of 100 and 200 mg/kg mitigated this decline in renal function, evidenced from the reduced levels of serum urea and creatinine. Limonene demonstrated anti-inflammatory effects by reducing pro-inflammatory cytokines (TNF-α, IL-1β, COX-2), suppressing the TLR4/NF-κB/AP-1 but not IRF3 signaling pathways, and modulating oxidative stress through Nrf2 activation. The results suggest that limonene holds promise as a potential therapeutic agent for mitigating intestinal inflammation and preserving gastrointestinal health.

## 1. Introduction

The intricate ecosystem of the human gastrointestinal system is crucial for nutrient absorption, immune function, and overall well-being. Within this complex network, the jejunum, a vital part of the small intestine, plays a pivotal role in the digestion and absorption of nutrients [1]. The intestinal barrier is a complex system that protects the body from harmful substances and pathogens in the intestinal lumen. It consists of several key elements, such as the mucus layer, the epithelial layer, the tight junctions, and the immune cells [2].

Inflammation is a fundamental physiological response orchestrated by the immune system to combat harmful stimuli, such as pathogens, injuries, or irritants. While inflammation is essential for tissue repair and host defense, dysregulated or chronic inflammation can lead to tissue damage and contribute to the pathogenesis of various inflammatory diseases, including inflammatory bowel disease (IBD), rheumatoid arthritis (RA), and asthma, among others. Consequently, there is a growing interest in identifying natural compounds with anti-inflammatory properties that can modulate the inflammatory response and offer therapeutic benefits [3].

In sepsis, the injury of the gut is a common pathophysiologic condition, which is proposed to be the main cause of serious illness due to the resultant malfunction of the intestinal barrier [4].

Lipopolysaccharide (LPS) is a molecule found in the outer membrane of Gram-negative bacteria. It enters the bloodstream and can trigger a strong immune response; therefore, it is known as an endotoxin [5]. Gut barrier dysfunction occurs when intestinal mucosal lesions occur, allowing the translocation of LPS and other bacterial products into the circulation [6]. This can cause inflammation, oxidative stress, and metabolic disorders, which may contribute to the development of various diseases, such as colorectal cancer (CRC), inflammatory bowel disease (IBD), obesity, diabetes, and liver disease [7]. LPS can contribute to gut barrier dysfunction via Toll-like receptor activation (TLR4) on the surface of epithelial cells and immune cells. When TLR4 interacts with a ligand, it forms homodimers. Two signaling pathways are initiated by this. First, from the plasma membrane, TLR4 induces the TIR domain-containing adapter protein (TIRAP) and myeloid differentiation primary response 88 (MyD88) pathway, which ultimately turns on NF-κB and activator protein 1 (AP-1) activation. Second, from the endosome, TLR4 induces the TIR-domain-containing adaptor protein inducing IFNβ (TRIF). Consequently, the TRIF-associated adaptor molecule (TRAM) pathway prompts the activation of interferon regulatory factor (IRF3), with subsequent production of type I interferons (IFNs) with a late wave of NF-κB stimulation [8].

The activation of the nuclear factor-kappa B (NF-κB) regulates the expression of pro-inflammatory cytokines, chemokines, and adhesion molecules [9]. These mediators can disrupt the tight junctions, increase the permeability of the epithelial layer, and recruit more immune cells to the site of inflammation. Moreover, LPS can also impair the function of intestinal alkaline phosphatase (IAP), an enzyme that detoxifies LPS in the lumen and reduces the production of mucus, which provides a physical barrier against bacteria [10].

Limonene, a monocyclic terpene hydrocarbon, is known for its diverse array of biological properties and has gained significant attention for its potential therapeutic applications in various fields [11]. The multifaceted nature of LPS-induced intestinal injury within the jejunum is a subject that has captivated the scientific community, as it can have far-reaching implications for gastrointestinal health and, by extension, systemic well-being. The ability of limonene to intervene in this process and mitigate the destructive effects of LPS is a promising avenue of research that could potentially lead to innovative preventive and therapeutic strategies [12]. A recent study showed that limonene can protect the gut barrier function in mice with colitis, an inflammatory bowel disease, by reducing inflammation, enhancing tight junctions, and boosting antioxidants in the colon [13]. Limonene has been used for the relief of heartburn and gastroesophageal reflux disorder because of its gastric-acid-neutralizing effect and improvement of peristalsis. Limonene has also been demonstrated to have a chemo-preventive activity against many types of cancer including gastric [14] and colorectal cancer [15]. Limonene also presents a gastroprotection effect against ethanol- and indomethacin-induced gastric ulcers [16].

Understanding the molecular mechanisms underlying the anti-inflammatory effects of limonene could provide valuable insights into its therapeutic potential for managing inflammatory disorders. Moreover, elucidating its mode of action may pave the way for the development of novel pharmacological interventions targeting NF-κB/AP-1 signaling in inflammatory conditions. Therefore, this study holds promise not only for advancing our understanding of inflammation but also for the development of new therapeutic strategies aimed at mitigating inflammatory diseases.

This article embarks on an exploration of the intriguing relationship between limonene and the prevention of LPS-induced intestinal injury in this crucial part of the gastrointestinal system. The present study aimed at investigating the anti-inflammatory effect of limonene against LPS-induced jejunal injury in mice.

## 2. Materials and Methods

### 2.1. Reagents

LPS (O55:B5) and limonene were purchased from Sigma Aldrich (Germany). Primers of TLR4, NF-κB, IRF3, AP-1, iNOS, IL-1β, TNF-α, COX-2, Nrf2, and GAPDH were purchased from Macrogen (Republic of Korea). The kits for urea and creatinine measurements were bought from Linear Chemicals (Spain).

### 2.2. Animal Experiments

This study was performed in the College of Pharmacy, University of Baghdad, in accordance with the guidelines for animal protocols approved by the Institutional Animal Care and Use Committee (approval number: FCB19722). Male mice with a body weight (BW) of 24 ± 3 g were raised in an animal facility under controlled conditions of temperature, humidity, and a 12 h light/dark cycle. The mice had free access to water and feed. Twenty-four albino male mice were randomly divided into four groups (6 in each group): a control group (CON; group 1), an administration group (LPS group; group 2), and two treatment groups, where the mice received limonene in two different doses in addition to LPS. The treatment groups included mice that received limonene at either 100 mg/kg (LM100 + LPS; group 3) or 200 mg/kg (LM200 + LPS; group 4). Limonene was administered orally by gavage daily in the morning for 5 consecutive days, while LPS was administered as a single IP injection (10 mg/kg) on day 5. Mice in the CON and LPS groups received equal volumes of water orally and daily for 5 days, treated exactly as the treatment groups; the CON group received an equal volume of phosphate-buffered saline IP injection on day 5 (instead of LPS). Euthanization was performed on day 6, exactly 24 h after the LPS injection; blood and tissue samples were then collected for analysis. Figure 1 represents a schematic illustration of the experimental animal design.

### 2.3. Blood Collection

Retro-orbital sampling was used to collect blood on day 6. Subsequently, the blood sample was allowed to coagulate at room temperature for 30 min. Following coagulation, the samples were centrifuged at 4 °C for 20 min at 3000 rpm. The resulting serum was carefully extracted using a micropipette and transferred into labeled micro-centrifuge tubes. These tubes were then stored at −20 °C to determine the urea and creatinine levels, which was performed 24 h after collection [9].

### 2.4. Biochemical Measurements

Serum urea and creatinine (Cr) levels were analyzed as pivotal indicators of renal injury severity. A semi-automated biochemical analyzer was used for the measurements by adhering to the guidelines provided by the manufacturer.

### 2.5. The Modified Murine Sepsis Score (MSS) System

The MSS system encompasses the assessment of seven components: appearance, level of consciousness, activity, response to stimulus, eyes, respiratory rate, and respiratory quality [17]. The traditional MSS score is derived by averaging these seven components for each group. This scoring system provides a valuable tool to assess the well-being of mice and to provide a measure of the general condition of mice. The MSS was determined by two observers; the first observer carried out the quantitative assessments before the LPS injection on day 5, and the next observer assessed 24 h after LPS injection, prior to euthanasia. One of the observers remained unaware of the treatment administered to ensure a rigorous MSS. Assessments were performed while the mice were inside their cages. Scores were documented and analyzed by averaging the scores of the 7 parameters for each group. In addition, the mortality rate was also reported for all groups to provide a further tool to assess the survival of animals and compare it to the treatment effectiveness.

### 2.6. Gene Expression Analysis

All animals were euthanized with pentobarbital (50 mg/kg) followed by cervical dislocation 24 h after LPS injection [18]. The proximal part of the jejunum was extracted and homogenized [19]. Gene expressions (mRNA) of TLR4, NF-κB, IRF3, AP-1, iNOS, IL-1β, TNF-α, COX-2, and Nrf2 were measured using qRT-PCR, with GAPDH used as a housekeeping gene [20,21,22]. Using the TransZol Up Plus RNA Kit (TransGen, biotech Beijing 100192, China), total RNA was isolated from the intestinal homogenate using TRIzol; thereafter, complementary DNA (cDNA) synthesis was carried out using the EasyScript^®^ one-step gDNA removal and cDNA synthesis (TransGen, biotech). SYBR Green Supermix was used to measure the mRNA expression levels (TransGen, biotech). The primer sequences were listed in Table 1, and they were created using IDT’s PrimerQuest program:

### 2.7. Statistical Analysis

The statistical analyses were carried out utilizing an ANOVA test followed by the Tukey post-test for multiple groups. To identify significant differences between all groups, *p* < 0.05 was used as the threshold for statistical significance. Analyses of data were performed with the Prism GraphPad 5 software.

## 3. Results

### 3.1. Effects of Limonene on Pro-Inflammatory Cytokines in LPS-Induced Intestinal Injury

Inflammatory markers were assessed to shed light on the molecular players in the event of intestinal inflammation induced by LPS. IL-1β is one of the important inflammatory markers, and it was measured in this study. Mice injected with LPS showed a significant elevation in their mRNA expression of IL-1β (33.39 ± 1.87- vs. 3.48 ± 1.54-folds) compared to the control group, suggesting an activated, ongoing inflammatory event in the jejunal tissue as shown in Figure 2A. Interestingly, treatment with limonene at 100 mg/kg and 200 mg/kg resulted in a significant attenuation of IL-1β expression levels (18.81 ± 3.14 and 12.28 ± 1.09 vs. 33.39 ± 1.87) compared to non-treated mice, respectively. The injection of LPS in mice resulted in a significant spike in TNF-α (Figure 2B), iNOS (Figure 2C) and COX-2 (Figure 2D) mRNA levels compared to the corresponding levels of the control mice: 35.35 ± 1.47 vs. 2.50 ± 0.96; 35.12 ± 6.99 vs. 1.73 ± 0.70; and 15.75 ± 2.43 vs. 1.38 ± 0.39. Moreover, the pretreatment of mice with limonene (both 100 mg/kg and 200 mg/kg) resulted in a significant decline in TNF-α gene expression (6.97 ± 1.14 and 2.81 ± 0.71 vs. 35.35 ± 1.47), compared to the non-treated LPS-induced mice. Consistently, the iNOS and COX-2 also showed significant downregulation (9.82 ± 3.48 and 9.41 ± 1.87 vs. 35.12 ± 6.99; 6.41 ± 1.14 and 3.32 ± 0.504 vs. 15.75 ± 2.43) after limonene treatment with either 100 mg/kg or 200 mg/kg, compared to the non-treated LPS-induced mice. This result indicates that limonene exerted a strong anti-inflammatory effect in jejunal inflammation induced by LPS.

### 3.2. Effects of Limonene on Inflammatory Pathways in LPS-Induced Intestinal Injury

The upstream inflammatory event was examined by measuring TLR4 (LPS receptor) mRNA levels (Figure 3A). The results showed that a significant upregulation of TLR4 was detected in LPS-challenged mice compared to the control group (1.71 ± 0.75- vs. 25.67 ± 4.16-fold), revealing an amplified inflammatory state. Interestingly, mice pretreated with either dose of limonene exhibited a significant downregulation of TLR4 (6.48 ± 1.27- and 2.90 ± 0.20- vs. 25.67 ± 4.16-fold) in jejunal tissue compared to non-treated animals.

Furthermore, our investigation was directed to the downstream pathways of TLR4 in the inflammatory cascade event that leads to the production of pro-inflammatory cytokines. Transcription factors from the Myd88-dependent and Myd88-independent pathways were assessed. In the Myd88-dependent pathway, NF-κB (Figure 3B) and AP-1 (Figure 3D) were studied to explore a more detailed view of the pathways targeted by limonene. In this signaling arm, limonene administration (100 and 200 mg/kg) resulted in a remarkable decline in gene expressions of NF-κB (Figure 3B) compared to non-treated mice (1.98 ± 0.95 and 0.48 ± 0.17 vs. 24.61 ± 4.22). In the same context, mice treated with limonene (100 and 200 mg/kg) showed a significant attenuation in jejunal AP-1 expression (Figure 3C) compared to non-treated mice (3.50 ± 1.079 and 1.89 ± 0.56 vs. 25.26 ± 8.78). In the Myd88-independent pathway, IRF3 was studied to explore a more detailed view of the pathways targeted by limonene; in contrast to the Myd88-dependent pathway, data showed that jejunal IRF3 mRNA (Figure 3C) did not change due to limonene treatment compared to non-treated mice, implying that limonene had no role in the Myd88-independent cascade post TLR4 activation.

### 3.3. Effect of Limonene on Oxidative Stress State in LPS-Induced Intestinal Injury

One of the critical regulators of the oxidative stress response is the transcription factor Nrf2. Our results revealed that limonene at 200 mg/kg significantly amplified Nrf2 gene expression in the jejunal tissues (Figure 4) compared to tissues from non-treated mice (18.15 ± 2.87 vs. 5.24 ± 0.308). In contrast, 100 mg/kg of limonene seemed to have an unnoticeable effect on Nrf2 expression. This result may suggest that limonene regulated the response of the cells to the oxidative stress by exerting antioxidant action that augmented its anti-inflammatory effects.

### 3.4. Effects of Limonene on Murine Sepsis Score (MMS) and Mortality Rate in LPS-Induced Intestinal Injury

In this study, the MSS was employed to assess the observed severity of the sepsis in each group. Figure 5A illustrates a significant increase in MSS in the LPS group compared to the CON group (2.57 ± 0.2 vs. 0.02 ± 0.02), while a significant reduction in the MSS was observed in groups receiving limonene at doses of 100 mg/kg and 200 mg/kg (0.8 ± 0.2 and 0.6 ± 0.2 vs. 2.57 ± 0.2) compared to the LPS group. These findings indicate that limonene reduced the severity of sepsis in mice. In addition to the MSS, the mortality was also reported in our study. As demonstrated in Figure 5B, a notable increase in the mortality rate (50%) was reported in the mice challenged with LPS, whereas a remarkable reduction was observed in the group receiving limonene at 100 mg/kg. Interestingly, there were no deaths reported in the group treated with 200 mg/kg of limonene. These results strongly augment the results from the measurement of other parameters in our study that limonene has the potential to mitigate the severity of sepsis in mice.

### 3.5. Effect of Limonene on Urea and Creatinine in LPS-Induced Intestinal Injury

Measuring urea and creatinine levels in LPS-induced jejunal injury in mice serves as a valuable approach to assess the impact of the injury and treatment on kidney function. Elevated levels of urea and creatinine in the serum are indicative of impaired kidney function, which can result from various factors, including sepsis-induced injury. The injection of LPS in mice has been demonstrated to induce renal function deterioration, as evidenced by a significant increase in serum urea and creatinine levels (serum urea: 116.35 ± 17.35 vs. 34.59 ± 3.55 mg/dL; creatine: 0.916 ± 0.02 vs. 0.26 ± 0.02 mg/dL) compared to those in the control mice (Figure 6A,B). Moreover, mice treated with limonene at doses of 100 mg/kg and 200 mg/kg had mitigated elevations in urea and creatinine and rescued renal function, as indicated by reduced levels of serum urea and creatinine (serum urea: 54.78 ± 3.31 and 36.26 ± 2.9 vs. 116.35 ± 17.35 mg/dL; creatine: 0.29 ± 0.02 and 0.21 ± 0.01 vs. 0.916 ± 0.02 mg/dL) compared to those observed in the LPS group.

## 4. Discussion

Due to the disruption of the intestinal barrier, gastrointestinal damage emerges as a prevalent pathophysiological concern in cases of sepsis, often regarded as the central driving force behind critical illness [23]. This phenomenon is characterized by a multifaceted interplay involving various factors such as heightened apoptosis, perturbed tight junctions, augmented cytokine production by the intestinal immune system, and intricated interactions with commensal bacteria and their byproducts. Despite the complex array of potential contributors, the precise mechanism underlying the development of intestinal mucosal lesions remains elusive [24].

In the context of our current investigations, we opted to employ a lipopolysaccharide (LPS)-induced model of intestinal injury. This experimental paradigm is widely recognized and utilized for its effectiveness in elucidating the intricate mechanisms that underlie inflammation specifically within the jejunum. By selecting this well-established model, we aimed to delve deeper into the complexities of jejunum inflammation, leveraging the robust foundation provided by previous research utilizing LPS-induced intestinal injury. [25,26]. Furthermore, the utilization of this model also afforded us the opportunity to explore and assess novel interventions aimed at ameliorating intestinal injury. By leveraging the versatility and reproducibility of the LPS-induced intestinal injury model, we were able to innovate and evaluate potential therapeutic strategies with the ultimate goal of mitigating the adverse effects associated with intestinal injury. This approach not only contributes to advancing our understanding of intestinal pathology but also holds promise for the development of innovative treatment modalities to address this clinical challenge.

The results of our animal experiments provide valuable insights into the potential health benefits of limonene, especially in the context of mitigating the adverse effects induced by LPS in mice. One notable observation from this study is that the mice exhibited signs of illness, including tiredness, lethargy, and overall unwell behavior, following the administration of LPS. These symptoms are indicative of the immune response triggered by LPS, which can include inflammation, fever, and a range of behavioral and physiological changes. What makes this study particularly interesting is the apparent contrast in the mice’s symptoms when limonene was introduced. The observed improvement in the mice’s well-being and health when they were given limonene suggests limonene as a potential agent to counteract the negative effects of LPS. In this study, the well-being of mice was assessed through measurements of the Murine Sepsis Score (MSS) and mortality rate. Consistent with previous research, mice injected with LPS showed elevated MSSs and increased mortality, indicating a deterioration in mouse health. This aligns with findings from other studies demonstrating that LPS administration induces a high MSS and increased mortality in mice [27,28].

Treatment with limonene at doses of 100 mg/kg and 200 mg/kg resulted in a decrease in the MSS and mortality rate compared to the LPS group. This suggests an improvement in the condition of mice suffering from intestinal inflammation induced by LPS and augment the findings reported from the measurement of other inflammatory markers. These results are further supported by the measurement of urea and creatinine levels, which were found to be elevated in the LPS group, indicating further renal injury, as reported in other study [29]. Furthermore, limonene administration led to a reduction in urea and creatinine levels, indicating a protective effect against renal injury. This observation is consistent with previous research demonstrating the potential therapeutic benefits of monoterpenes in mitigating sepsis-induced organ dysfunction [30].

The results of our studies also provide a comprehensive view of the potential therapeutic benefits of limonene in mitigating the inflammatory and oxidative stress responses induced by LPS in mice. There are several key points highlighted from our studies. Limonene treatment effectively reduced the production of inflammatory cytokines, such as TNF-α, IL-1β, and COX-2, compared to the mice that received LPS alone. These findings align with previous studies showcasing the anti-inflammatory properties of limonene in various animal models [31,32]. TNF-α is a one of the cytokines that cause the activation and recruitment of inflammatory cells, which eventually lead to the activation of the vascular endothelium, nitric oxide release, and, hence, local vasodilation, and increase vascular permeability [33,34]. IL-1β, another cytokine, causes increases in the production of the adhesion molecules on endothelial cells, such as intercellular adhesion molecule 1 (ICAM-1) and vascular cell adhesion molecule 1 (VCAM-1) [35]. COX-2 is an enzyme that plays a significant role in inflammation and pain within the body [36].

LPS injection in group 2 (LPS only) showed a significant increase in TNF-α, IL-1β, and COX-2 levels, which is consistent with previous studies [37,38]. On the other hand, limonene treatment in groups 3 and 4 (LM100 + LPS and LM200 + LPS, respectively) reduced the production of TNF-α, IL-1β, and COX-2 compared to the mice that received LPS injection in group 2. Our result is consistent with previous studies that showed the anti-inflammatory effect of limonene in different animal models [39,40]. Furthermore, there are other studies that discuss the anti-inflammatory effects of other terpenes in different experimental models [41].

Nitric oxide (NO) is produced by the enzyme iNOS (inducible nitric oxide synthase) in response to inflammation or infection. Normally, NO can act as a vasodilator; however, excessive NO production in the intestines could result in oxidative stress and tissue damage [42]. LPS-induced intestinal injury is accompanied by increased production of NO, which results from the cytokine-mediated upregulation of iNOS [43]. Therefore, during sepsis, intestinal function is reduced, causing systemic vasodilatation. For these reasons, the selective inhibition of iNOS, leaving physiologically present eNOS intact, might be a good approach for the treatment of sepsis-induced intestinal injury [44]. When group 2 received 10 mg/kg of LPS, their iNOS expression increased significantly. However, 100 mg/kg or 200 mg/kg of limonene reduced the mRNA expression of iNOS levels significantly. This shows that limonene has an anti-inflammatory effect and can prevent the excessive production of NO caused by LPS, as previous studies have also indicated [45,46].

In our study, the mRNA expression of TLR4, NF-ĸB, AP-1, and IRF3 were measured to evaluate the effect of limonene on the MyD88-dependent and MyD88-independent signaling pathways. We showed notable results regarding the mRNA expression of TLR4, NF-κB, AP-1, and IRF3, which show significant upregulation in the LPS group in comparison to the normal group. These findings align with those of Hu and colleagues, who revealed that LPS injection in a mouse model induced the gene expression of TLR4, MyD88, and NF-κB in the intestinal tissue at different time intervals, with maximum upregulation being detected after 24 h. Furthermore, another study also demonstrates that LPS injection induces the upregulation of TLR4 [47].

Our results also showed that limonene has a significant anti-inflammatory effect, since pretreatment with 100 mg/kg and 200 mg/kg of limonene daily for five days significantly attenuated the gene expression of TLR4, NF-ĸB, and AP-1 in jejunal tissues. As documented by many studies, limonene was investigated previously for its anti-inflammatory effect in an animal model by decreasing NF-κB expression [48,49]. On the other hand, the pretreatment of mice with low and high doses of limonene showed no significant attenuation in IRF3 levels compared to the LPS group. These findings demonstrate that limonene action focuses on the Myd88-dependent pathway of the inflammatory cascade event and produces a strong anti-inflammatory effect, targeting the TLR4/AP-1/NF-κB axis but not IRF3.

In contrast to another study, involving intraperitoneal injections of 10 mg/kg of LPS in mice, the LPS group had higher NF-κB nuclear translocation in the jejunum but no significant difference in Nrf2 nuclear translocation [50]. In addition, another study revealed that Nrf2-related antioxidant genes did not upregulate following LPS challenge [51]. On the other hand, it appears that the administration of 200 mg/kg of limonene resulted in a significant increase in Nrf2 levels compared to group 2, while the 100 mg/kg dose did not show any significant differences. The results suggest a dose-dependent effect, indicating that the increase in Nrf2 levels is more pronounced with the higher dose of 200 mg/kg, and this provides further evidence of the intestinal-protective effect of limonene.

A previous study also showed that limonene protects skin keratinocytes from the damaging effect of UVB radiation. This protection occurs by activating the Nrf2-dependent antioxidant defense system, which helps the skin cells to defend themselves against oxidative stress and damage caused by UVB exposure [52].

## 5. Conclusions

In conclusion, limonene exerts strong anti-inflammatory effects, targeting inflammatory TLR4/AP1/NF-κB signaling pathways. Our animal experiments shed light on the promising role of limonene in ameliorating the effects of LPS-induced intestinal inflammation in mice. As discovery of effective therapies for sepsis-related conditions is of crucial importance, limonene offers a reliable solution. Further studies could be conducted to explore a more detailed effect of limonene on inflammatory events and to investigate the effects of limonene on various organs and different sepsis models. In addition, toxicological studies of limonene are also important. Nonetheless, our studies highlight the importance of natural compounds like limonene as potential therapeutic agents for conditions involving excessive inflammation and immune system activation.

## Figures and Tables

**Figure 1 biomolecules-14-00334-f001:**
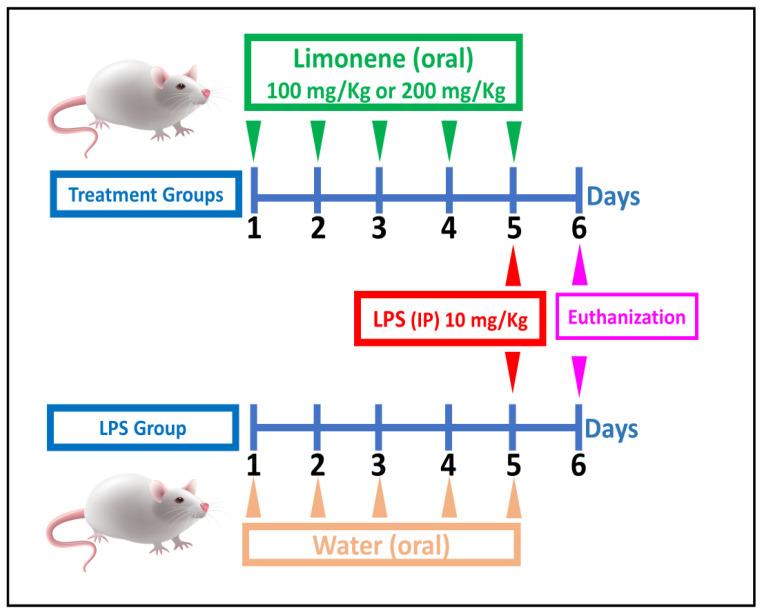
Schematic presentation of experimental animal design.

**Figure 2 biomolecules-14-00334-f002:**
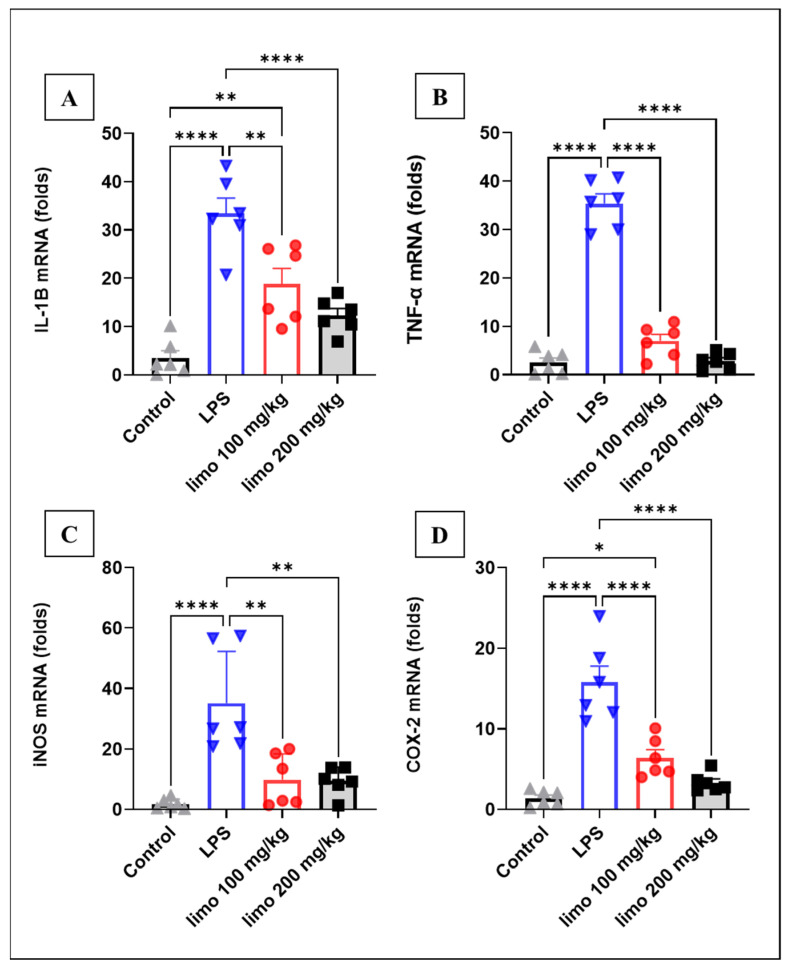
Effects of limonene on pro-inflammatory cytokines in LPS-induced intestinal injury in mice. Data represent mean ± SEM for mRNA expression of pro-inflammatory cytokines measured 24 h after induction with LPS (10 mg/kg) in mice jejunal tissue: (**A**) IL-1β; (**B**) TNF-α; (**C**) iNOS; (**D**) COX-2. Limonene treatment used in 2 doses (100 mg/Kg and 200 mg/kg) for 5 consecutive days before LPS injection. Calculations of gene expression performed relative to GAPDH as a control gene. n = 6 in each group. Analysis of data was performed with Prism GraphPad 5. * *p* < 0.05; ** *p* < 0.01; **** *p* < 0.0001 indicate statistical significance. Control: 
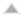
; LPS: 
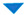
; limonene 100 mg/kg: 
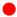
; limonene 200 mg/kg: 
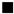
.

**Figure 3 biomolecules-14-00334-f003:**
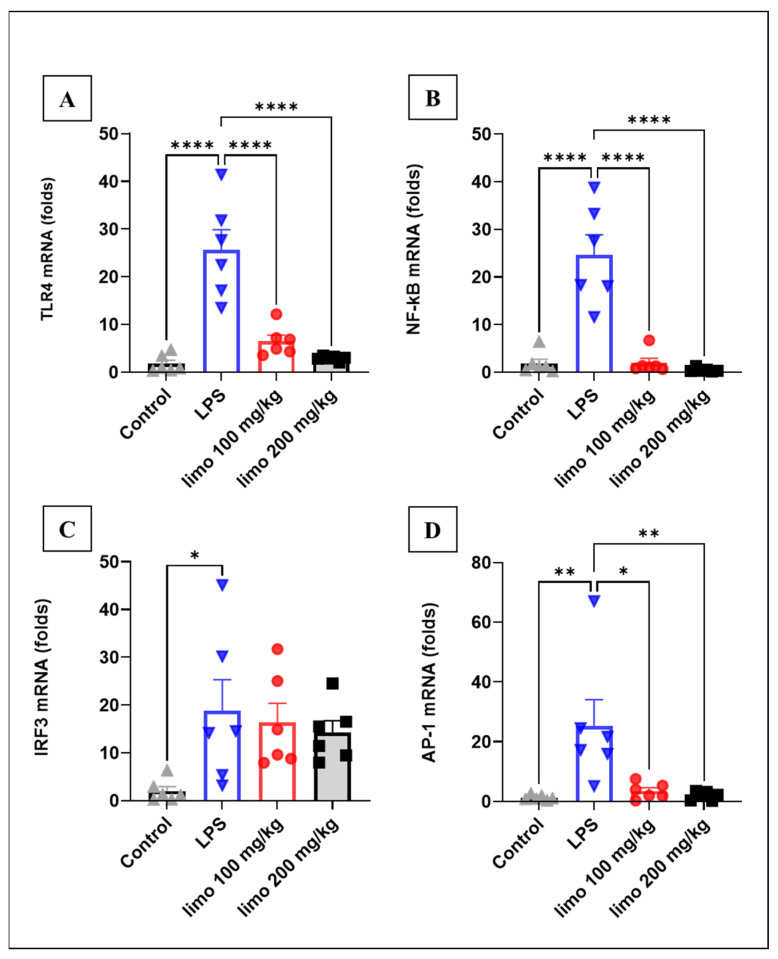
Effects of limonene on inflammatory pathways in LPS-induced intestinal injury in mice. Data represent mean ± SEM for mRNA expression of inflammatory markers measured 24 h after induction with LPS (10 mg/kg) in mice jejunal tissue: (**A**) TLR4; (**B**); NF-κB; (**C**) IRF3; (**D**) AP-1. Limonene treatment used in 2 doses of 100 mg/kg and 200 mg/kg for 5 consecutive days before LPS injection. Calculations of gene expression performed relative to GAPDH as a control gene. n = 6 in each group. Analysis of data was performed with Prism GraphPad 5. * *p* < 0.05; ** *p* < 0.01; **** *p* < 0.0001 indicate statistical significance. Control: 
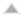
; LPS: 
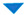
; limonene 100 mg/kg: 
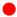
; limonene 200 mg/kg: 
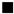
.

**Figure 4 biomolecules-14-00334-f004:**
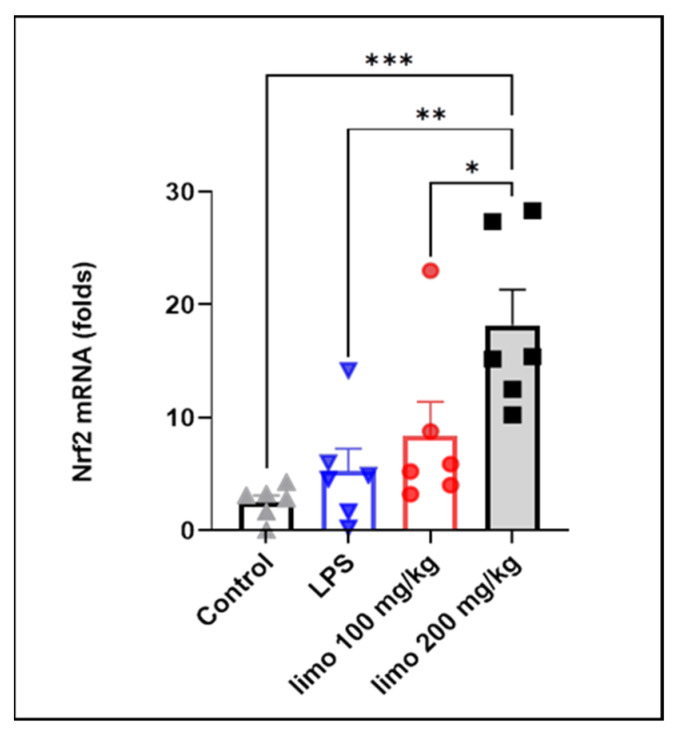
Effect of limonene on the jejunal expression of Nrf2. Data represent mean ± SEM for mRNA expression of Nrf2 measured 24 h after induction with LPS (10 mg/kg) in mice jejunal tissue. Limonene treatment used in 2 doses (100 mg/kg and 200 mg/kg) for 5 consecutive days before LPS injection. Calculations of gene expression performed relative to GAPDH as a control gene. n = 6 in each group. Analysis of data was performed with Prism GraphPad 5. * *p* < 0.05; ** *p* < 0.01; *** *p* < 0.001 indicate statistical significance. Control: 
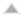
; LPS: 
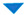
; limonene 100 mg/kg: 
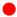
; limonene 200 mg/kg: 
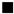
.

**Figure 5 biomolecules-14-00334-f005:**
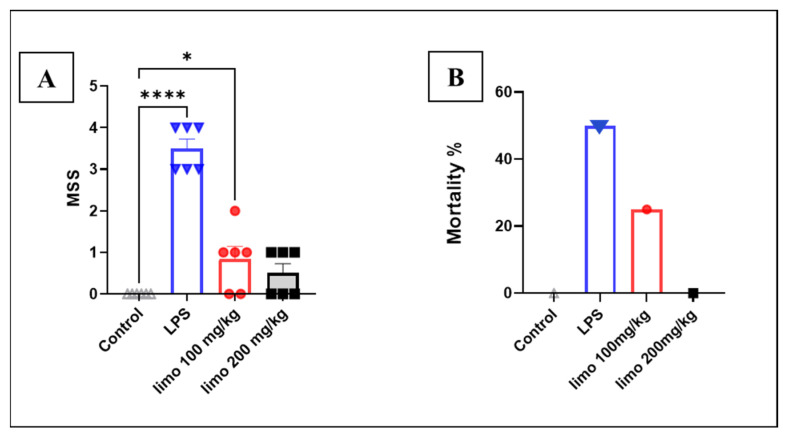
Effect of limonene on mice mortality and Murine Sepsis Score (MMS) in LPS-induced intestinal injury. Data represented as mean ± SEM for Murine Sepsis Score (MMS) and percentage for mortality. (**A**) MSS; (**B**) mortality. Limonene treatment used in 2 doses (100 mg/Kg and 200 mg/kg) for 5 consecutive days before LPS injection. n = 6 in each group. Analysis of data was performed with Prism GraphPad 5. * *p* < 0.05; **** *p* < 0.0001 indicate statistical significance. Control: 
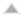
; LPS: 
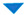
; limonene 100 mg/kg: 
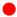
; limonene 200 mg/kg: 
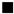
.

**Figure 6 biomolecules-14-00334-f006:**
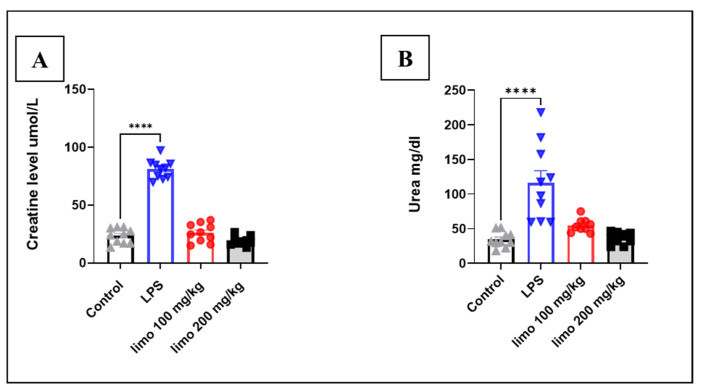
Effect of limonene on urea and creatinine in LPS-induced intestinal injury. Data represent mean ± SEM for urea and creatinine measured 24 h after induction with LPS (10 mg/kg) in mice jejunal tissue: (**A**) urea; (**B**) creatinine. Limonene treatment used in 2 doses (100 mg/Kg and 200 mg/kg) for 5 consecutive days before LPS injection. n = 6 in each group. Analysis of data was performed with Prism GraphPad 5. **** *p* < 0.0001 indicate statistical significance. Control: 
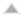
; LPS: 
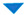
; limonene 100 mg/kg: 
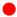
; limonene 200 mg/kg: 
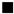
.

**Table 1 biomolecules-14-00334-t001:** Primer sequences.

Primers	Sequences 5′→3′ Direction
GAPDH	Forward: CGGGTTCCTATAAATACGGACTG Reverse: CCAATACGGCCAAATCCGTTC
TLR4	Forward: TCCCTGCATAGAGGTAGTTCC Reverse: TCAAGGGGTTGAAGCTCAGA
NF-κB	Forward: AAGACAAGGAGCAGGACATG Reverse: AGCAACATCTTCACATCCC
IRF3	Forward: CAATTCCTCCCCTGGCTAGA Reverse: GGGATCCTGAACCTCGTTCG
AP-1	Forward: GCTGCAGGATGATGCGATAG Reverse: TTCTAGCCAGGACGACTTGC
iNOS	Forward: GGTGAAGGGACTGAGCTGTT Reverse: ACGTTCTCCGTTCTCTTGCAG
IL-1β	Forward: TGCCACCTTTTGACAGTGATG Reverse: TGATGTGCTGCTGCGAGATT
TNF-α	Forward: TAGCCCACGTCGTAGCAAAC Reverse: ACAAGGTACAACCCATCGGC
COX-2	Forward: GCTCAGCCAGGCAGCAAATC Reverse: CACCATAGAATCCAGTCCGGG

## Data Availability

Dataset available on request from the authors.

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
