# Peer review of "Limonene Exerts Anti-Inflammatory Effect on LPS-Induced Jejunal Injury in Mice by Inhibiting NF-κB/AP-1 Pathway"

_biomolecules, 2024, doi:10.3390/biom14030334_

Round 1

Reviewer 1 Report

Comments and Suggestions for Authors

The study suggests that limonene holds promise as a potential therapeutic agent for mitigating intestinal inflammation and preserving gastrointestinal health. However, several points require further clarification and additional support:

a. The expression of cytokines was analyzed after treatment with limonene in mice with induced GI injuries caused by LPS. Could you provide more details on which specific pro-inflammatory mechanisms were reduced by limonene?

The study mentions the treatment of LPS causing inflammation not only in the jejunum but throughout the entire gastrointestinal tract. Could you clarify whether the observed reduction in pro-inflammation applies to other parts of the GI tract as well?

Is there any observed change in phenotype, such as Disease Activity Index (DAI) or histological score, after treatment with limonene? Additional information on these parameters would strengthen the comprehensiveness of the findings.

Author Response

Reviewer#1

The study suggests that limonene holds promise as a potential therapeutic agent for mitigating intestinal inflammation and preserving gastrointestinal health. However, several points require further clarification and additional support:  

Response:  Thank you very much for your constructive comments.  Below are our point-by-point responses to your comments.

  1. The expression of cytokines was analyzed after treatment with limonene in mice with induced GI injuries caused by LPS. Could you provide more details on which specific pro-inflammatory mechanisms were reduced by limonene?

Response:  LPS induces inflammation through its receptor TLR4. Activation of TLR4 triggers 2 main signaling pathways.  Each pathway involves a complex of molecular players; however, there are important key signaling molecules that can be measured to provide a conclusion on which pathway is involved. 

The first pathway involves Myd88-dependent pathway where the TLR4-LPS interaction will remain on surface membrane.  We assessed NFkB and AP-1 as markers of the Myd88-dependent pathway, as these signaling molecules are crucial signaling molecules and are considered the downstream transcription factors, prior to membrane TLR4 signaling pathways.  The NFkB and AP-1 also responsible for activation of transcription of proinflammatory cytokines and other components of inflammation.

The second pathway involves Myd88-independent pathway where the TLR4-LPS complex will be internalized via endosome that will trigger another signaling cascade independent of Myd88.  IRF3 was assessed, because it is the hallmark of Myd88-independent pathway as it is one of the important downstream molecules placed at the end of this pathway, i.e. post to TLR4 endosome signaling pathway. 

According to results obtained from our study, we could conclude that limonene exerts its effects on the Myd88-dependent pathway as it reduced the markers of this pathway (NFkB and AP-1) while not through Myd88-independent pathway as IRF3 were not affected by limonene (Fig.3).

The study mentions the treatment of LPS causing inflammation not only in the jejunum but throughout the entire gastrointestinal tract. Could you clarify whether the observed reduction in pro-inflammation applies to other parts of the GI tract as well?  

Response:  LPS is a well-known endotoxin that causes multi-organ inflammation including gastrointestinal tract (GIT).  LPS can induce injury along the GIT including duodenum, jejunum and colon. In our study, we investigated the effects of limonene on jejunum only, and other parts of GIT like duodenum and colon for example were not investigated. However, as long as LPS induces injury in various organs and various parts of GIT by a specific known mechanism that involves TLR4 activation and its downstream signaling pathways, it is expected that limonene may produce anti-inflammatory effects and mitigate injury in other GIT parts as well.  Limonene could also produce anti-inflammatory effects in other GIT parts, similar to the effect of limonene on jejunum. This is an interesting topic for future research to focus on other parts of GIT.

Is there any observed change in phenotype, such as Disease Activity Index (DAI) or histological score, after treatment with limonene? Additional information on these parameters would strengthen the comprehensiveness of the findings.

Response:  Although jejunal tissue were obtained from all animals during experiments, the tissues were all used for signaling studies; the histopathological examination were therefore not performed. However, during the experiments, we documented animal conditions and well-being according to murine sepsis score (MSS) scoring system.  We have now included the MSS data in this revision (new Fig.5a). In addition to that, we have also included the mortality rate of each group (new Fig.5b) as it is also considered an important parameter that strengthens our results. In addition, we also added serum creatinine and urea measurements for these mice to the results (new Fig.6).  Together, these additional results demonstrated observed changes in phenotype to strengthen the comprehensiveness of our fundings.  They also provided information on the well-being of the mice treated with limonene compared to nontreated group.

Reviewer 2 Report

Comments and Suggestions for Authors

The present study titled "Limonene exerts anti-inflammatory effect on LPS-induced jejunal injury in mice by inhibiting NF-κB/AP-1 pathway" by Kathem et al. is particularly interesting. In this research, the authors explored the protective properties of limonene, a natural compound known for its diverse biological effects, against LPS-induced injury in the jejunum of mice. The findings indicate that limonene shows anti-inflammatory effects by reducing levels of proinflammatory cytokines (TNF-α, IL-1β, COX-2), suppressing the TLR4/NF-κB/AP-1 pathways (while not affecting IRF3 signaling), and regulating oxidative stress through the activation of Nrf2. Therefore, the researchers propose that limonene holds great potential as a therapeutic agent for alleviating intestinal inflammation and maintaining gastrointestinal well-being.

Strengths:

-The quality of the writing is good, and the general structure of the manuscript is well-structured.

-Figures presented in the results section are well-structured.

Limitations:

- More details should be given in the materials and methods section.

- A summarized figure should be more valuable to summarize "Animal Experiment Design Section".

- The style of references used in the text should be checked (e.g. Line 34; [2] [3] or [2-3]).

- Units of numerical data should be checked (e.g. Line 189, 197; 100mg/kg or 100 mg/kg).

- The conclusion section should be added to highlight future remarks, importance, and relevance of their work.

Author Response

Reviewer#2

Strengths: 

-The quality of the writing is good, and the general structure of the manuscript is well-structured. 

-Figures presented in the results section are well-structured. 

Response:  We truly appreciate such encouraging comments from the Reviewer.  Our point-by-point responses to your comments are shown below.

Limitations: 

- More details should be given in the materials and methods section. 

Response:  More details of the methodology have now been provided in the Materials and Methods section.  Additional methods are shown in blue color texts.

- A summarized figure should be more valuable to summarize "Animal Experiment Design Section". 

Response:  As requested, we have now provided a new figure to summarize our Study Design (new Fig.1).

- The style of references used in the text should be checked (e.g. Line 34; [2] [3] or [2-3]). 

Response:  Thank you for bringing this up to our attention.  The style of the references has now been corrected.

- Units of numerical data should be checked (e.g. Line 189, 197; 100mg/kg or 100 mg/kg). 

Response:  We appreciate the attention to the details provided by the Reviewer.  We have now corrected the units accordingly.

- The conclusion section should be added to highlight future remarks, importance, and relevance of their work.

Response:  As requested, we have now added the future remarks, importance and relevance of our studies.  These additions are shown in the blue color texts in the Discussion section.  Thank you for your suggestions.

Reviewer 3 Report

Comments and Suggestions for Authors

Limonene exerts anti-inflammatory effect on LPS- induced jejunal injury in mice by inhibiting NF-κB/AP-1 pathway, uses mRNA induction by LSP in the jejunum of mice and rescue by limonene to suggest the use of limonene in GI inflammation.  The methodology is one-dimensional resulting in over interpretation of the outcome.

Specifically, the authors offer up information on the wellbeing of the mice as a result of LPS injection and treatment with limonene in the Discussion section (lines 216-217).  This should be presented in the Results section along with some documentation of the symptoms.  Second, there is room in Figure 3 for additional mRNA results based on the placement of 4 panels in Figures 1 and 2.  Induction of Nrf2 mRNA is not a good standard for its action.  Given your choice of mRNA levels as the primary data, you should look at genes regulated down-stream of NRF2, such as Hmox1, Gclm, Gclc, and possibly, Gpx2. All are considered antioxidant genes.   Gpx2 levels might be too low to measure in jejunum.

Comments on the Quality of English Language

OK

Author Response

Reviewer#3

Limonene exerts anti-inflammatory effect on LPS- induced jejunal injury in mice by inhibiting NF-κB/AP-1 pathway, uses mRNA induction by LSP in the jejunum of mice and rescue by limonene to suggest the use of limonene in GI inflammation.  The methodology is one-dimensional resulting in over interpretation of the outcome.  Specifically, the authors offer up information on the wellbeing of the mice as a result of LPS injection and treatment with limonene in the Discussion section (lines 216-217).  This should be presented in the Results section along with some documentation of the symptoms. 

Response:  Thank you for your suggestion.  As per your request for additional information of well-being of mice documented in the study, we have now provided the MSS scores (murine sepsis score) and mortality rate for data (new Fig.5). In addition, we have also added serum levels of urea and creatinine collected in this study (new Fig.6). All these parameters will provide hopefully strengthen the interpretation of our results and the well-being of mice from the limonene treatment.  To clarify why we did not present these data at the first time, we intended to publish another article of a related subject, and we tried to preserve these data for that purpose. Nonetheless, we have now added these data to the revised manuscript.

Second, there is room in Figure 3 for additional mRNA results based on the placement of 4 panels in Figures 1 and 2.  Induction of Nrf2 mRNA is not a good standard for its action.  Given your choice of mRNA levels as the primary data, you should look at genes regulated down-stream of NRF2, such as Hmox1, Gclm, Gclc, and possibly, Gpx2. All are considered antioxidant genes.   Gpx2 levels might be too low to measure in jejunum.

Response:  We are thankful for your valuable comment on Nrf2.  Many published studies indeed use this gene as a marker for antioxidant state of the cells.  Unfortunately, we could not perform additional measurements of the downstream genes of Nrf2, because we do not have any samples preserved.  Of note is that our animal studies were performed over 2 years ago.  There had been a lot of unforeseeable delays in completing the study and preparing the manuscript.  At this time, while we understand that the downstream of Nrf2 is important, we hope that measurement of Nrf2 can at minimum provide a window of opportunity to predict the downstream effect of Nrf2 by limonene.  We thank you for understanding our situation.

Round 2

Reviewer 1 Report

Comments and Suggestions for Authors

The manuscript has addressed all of my concerns but I still have minor comments. In Figure 2-6, there is an issue with the X-axis labels: there is no space between the number and unit (e.g., 'limo 100mg/kg'), but it should have a space between the number and unit. (e.g., 'limo 100 mg/kg')

 to use no space between a number and unit

Author Response

Thank you for bringing this up to our attention, we appreciate the attention to the details provided by the Reviewer. The issue of the space between limo and numbers corrected in all figures. Please see the attachement

Reviewer 3 Report

Comments and Suggestions for Authors

The revision is fine for publication

Author Response

Thank you for your valuable comment